# Cure-SFT: Diagnostic-Guided Data Curation for Instruction Tuning

Yuankang Fu[1]  Xinrong Gong[2]  Chen Gong[3]  Tong Zhang[1]  Kaixiang Yang[1]

## Abstract

Instruction data curation is central to improving the instruction-following ability of large language models. However, existing approaches often struggle to simultaneously maintain data quality, diversity, and distributional consistency, largely because they do not explicitly distinguish semantic redundancy from quality defects and rely on coarse-grained modeling of instruction data quality. To address this issue, we propose Cure-SFT, a coarse-to-fine, diagnostic-guided method for instruction data curation that explicitly disentangles semantic redundancy from quality defects. Specifically, Cure-SFT removes redundant samples via stratified semantic-geometric sampling, applies teacher models for diagnostic triage, and performs targeted defect remediation on fixable samples. Our experiments show that Cure-SFT can surpass full-data instruction tuning using only 10% of the data budget. Moreover, Cure-SFT consistently outperforms strong selection-based and rewriting-based baselines across data budgets, supporting the effectiveness of diagnostic-guided data curation. Our code is available at https://github.com/As1yk/Cure-SFT.

## 1. Introduction

Instruction tuning has become one of the common approaches for improving the instruction-following behavior of large language models (LLMs) (Ouyang et al., 2022; Zhang et al., 2026). A number of recent studies suggest that the effectiveness of instruction tuning does not merely depend on data scale, but is largely shaped by the overall quality of the instruction data (Zhou et al., 2023; Luo et al.,

2025). This observation has motivated further investigation into the curation of instruction data (Cao et al., 2024; Liu et al., 2024a; Chen et al., 2025). Despite recent progress in instruction data curation, existing approaches still lack a systematic solution that can jointly balance data quality, diversity, and distributional consistency under constrained data budgets.

In the context of instruction data curation, existing studies can be broadly categorized into two representative paradigms: Selection-based approaches (e.g., AlpaGasus (Chen et al., 2024b), IFD (Li et al., 2024b), and Deita (Liu et al., 2024b)) typically rely on external model scores or heuristic criteria to rank instruction data and discard low-scoring samples; whereas rewriting-based approaches (e.g., Alpaca-GPT4 (Peng et al., 2023) and CoachLM (Liu et al., 2024c)) leverage stronger teacher models to rewrite the original samples. Although both paradigms have improved model performance to some extent, they still face two unresolved challenges shared at a deeper level.

First, existing approaches fail to distinguish two orthogonal issues in instruction data: semantic redundancy and quality defects. Consequently, they tend to adopt coarse-grained, global strategies that uniformly remove or rewrite samples (Chen et al., 2024b; Cao et al., 2024; Peng et al., 2023). Such coarse-grained treatments fail to differentiate between frequent but semantically simple instructions and samples that are semantically distinctive yet contain localized quality defects (Lu et al., 2024). Consequently, semantic redundancy is not effectively reduced, while potentially high-value samples are often discarded due to minor quality issues.

Second, existing paradigms often implicitly assume a binary view of data quality—"usable" versus "unusable" (Chen et al., 2024b; Du et al., 2023). This assumption overlooks a more realistic three-state structure, consisting of high-quality samples, useless samples, and samples that are defective yet fixable. Without explicitly characterizing fixable samples, selection-based methods tend to discard them outright, despite their retained semantic value. Rewriting-based methods, by contrast, apply indiscriminate intervention across samples of different quality states, which may introduce unnecessary modifications to already high-quality data, leading to style homogenization and distributional

[1]School of Computer Science and Engineering, South China University of Technology, Guangzhou, China. [2]School of Engineering, Huaqiao University, Quanzhou, China. [3] School of Automation and Intelligent Sensing, Shanghai Jiao Tong University, Shanghai, China. Correspondence to: Kaixiang Yang <yangkx@scut.edu.cn>.

*Proceedings of the 43rd International Conference on Machine Learning*, Seoul, South Korea. PMLR 306, 2026. Copyright 2026 by the author(s).

shifts (Gudibande et al., 2024). Together, these issues call for an instruction data curation approach that explicitly separates semantic redundancy from quality defects and models the multi-state nature of data quality.

Based on the above analysis, we propose Cure-SFT, a diagnostic-guided instruction data curation method that follows a coarse-to-fine paradigm. Cure-SFT first applies a stratified semantic-geometric sampling strategy in the embedding space to remove semantically redundant samples while preserving broad semantic coverage. Subsequently, a teacher model performs diagnostic triage on the remaining samples, categorizing them into three groups: *High Quality*, *Fixable*, and *Useless*. Cure-SFT then adopts a diagnostic-guided targeted remediation strategy to selectively remediate *Fixable* samples while retaining high-quality samples in their original form. Through these design choices, Cure-SFT improves overall data quality while preserving data diversity and distributional consistency.

We conduct systematic instruction-tuning experiments on public datasets, including Alpaca and WizardLM, and evaluate Cure-SFT across a range of benchmark tasks. The results show that, under the same data budget, Cure-SFT consistently outperforms a variety of baseline methods. Our main contributions are summarized as follows: (1) We revisit instruction data curation from the perspective of disentangling semantic redundancy and quality defects. Existing selection- and rewriting-based pipelines often treat these factors in a coupled manner and underexploit fixable samples that retain semantic value. (2) We propose Cure-SFT, a coarse-to-fine diagnostic-guided data curation method that mitigates semantic redundancy via stratified semantic-geometric sampling, performs diagnostic triage using a teacher model, and applies diagnostic-guided targeted remediation to fixable samples. (3) Extensive experiments across multiple instruction-tuning benchmarks demonstrate that Cure-SFT achieves consistent gains over strong baselines under different data budgets, while largely preserving data diversity and distributional consistency.

## 2. Related Work

We review prior work on instruction data curation for instruction tuning, with a focus on selection-based and rewriting-based approaches.

**Instruction Data Selection** Instruction data selection methods aim to identify subsets of instruction data with higher training utility from large candidate pools under limited data budgets (Zhang et al., 2025). Representative approaches such as IFD (Li et al., 2024b), Deita (Liu et al., 2024b), and AlpaGasus (Chen et al., 2024b) typically assign scores to instruction samples using model feedback, heuristic criteria, or external evaluators, and retain top-ranked samples

for training. These methods have shown effectiveness in improving average data quality; However, their design primarily focuses on filtering data at the sample level, where instruction instances are either retained or discarded. As a result, semantically distinctive long-tail samples that contain only localized defects are often removed rather than refined, which may limit semantic coverage and diversity in the curated dataset (Chen et al., 2024b; Lu et al., 2024).

**Rewriting-Based Data Curation** Another line of work improves instruction data quality through rewriting or regeneration using stronger teacher models. Approaches such as Alpaca-GPT4 (Peng et al., 2023) and CoachLM (Liu et al., 2024c) leverage large language models to rewrite instruction–response pairs, aiming to globally enhance response quality. Compared to selection-based methods, rewriting provides a more direct mechanism for improving imperfect samples. Existing methods typically adopt a holistic rewriting strategy, applying uniform modifications across samples without explicitly distinguishing between different data quality states. While effective for large-scale quality enhancement, such uniform rewriting may introduce unnecessary changes to already high-quality samples, potentially leading to stylistic homogenization or distributional shifts (Gudibande et al., 2024; Shumailov et al., 2024).

## 3. Methodology

### 3.1. Method Overview

Figure 1 provides an overview of Cure-SFT. Cure-SFT explicitly decouples semantic redundancy from quality defects and addresses them through a three-stage pipeline. In Stage I, semantically redundant samples are removed in the embedding space to construct a candidate dataset with broad semantic coverage. In Stage II, a teacher model is used to perform fine-grained quality diagnosis, categorizing samples into three groups: *High Quality*, *Fixable*, and *Useless*. In Stage III, Diagnostic-Guided Targeted Remediation is applied to selectively correct and refine *Fixable* samples, which are then combined with high-quality samples preserved in their original form to form the final training set.

### 3.2. Stage I: Stratified Semantic-Geometric Sampling

Stage I is designed to mitigate semantic redundancy by constructing a compact candidate set that preserves broad semantic coverage under a fixed data budget $B$. Specifically, we operate in the embedding space (Abbas et al., 2023) and adopt a stratified sampling strategy that combines global semantic clustering with local geometric selection.

**Global Semantic Clustering.** Formally, let $D = \{x_i\}_{i=1}^{N}$ denote the initial dataset, where each sample $x_i = (I_i, C_i, R_i)$ consists of an instruction, an optional input context, and a corresponding response. We construct a unified

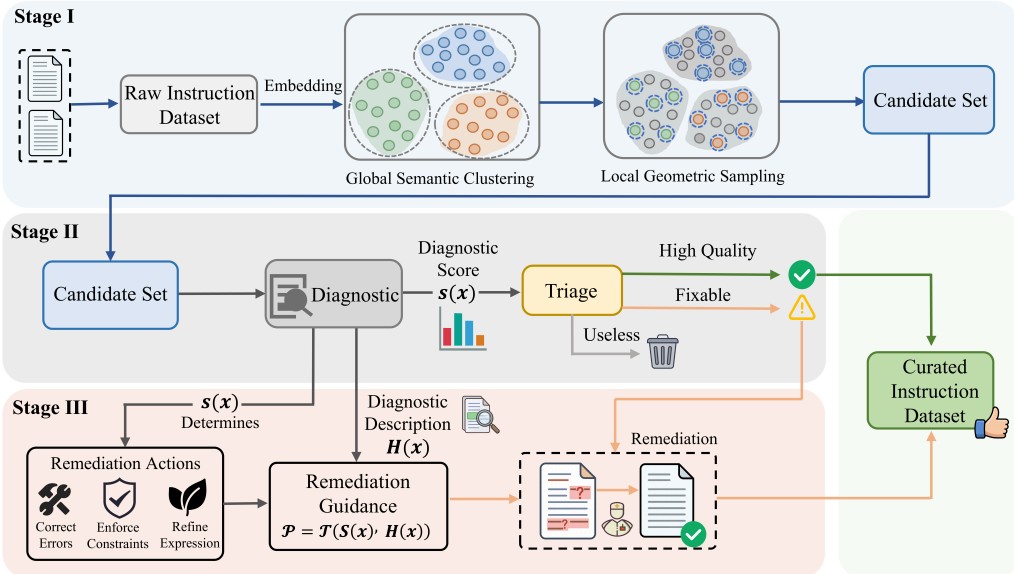

*Figure 1.* Overview of the proposed Cure-SFT. The pipeline consists of three sequential stages: (I) Stratified Semantic-Geometric Sampling, which selects a semantically diverse candidate set from the raw instruction dataset; (II) Diagnostic Triage, where a teacher model categorizes samples into *High-Quality*, *Fixable*, or *Useless* subsets; and (III) Diagnostic-Guided Targeted Remediation, which utilizes multi-dimensional diagnostic scores and diagnostic description to identify specific flaws and guide the remediation of *Fixable* samples.

query representation $q_i = I_i \oplus C_i$ and map it into a continuous semantic space using a pretrained encoder (Reimers & Gurevych, 2019):

$$v_i = f_\theta(q_i) = f_\theta(I_i \oplus C_i), \qquad (1)$$

where $\oplus$ denotes concatenation, and $v_i$ represents the resulting sentence embedding. In this embedding space, samples with similar semantic intent tend to exhibit strong geometric proximity, making distance-based clustering feasible (Mikolov et al., 2013). We then apply K-means to partition the embeddings into $K$ clusters $C_1, \ldots, C_K$, where $K = \lceil \sqrt{N} \rceil$ and $N$ denotes the total number of samples in the dataset, allowing the number of clusters to scale naturally with the dataset size.

**Local Geometric Sampling.** Given a total data budget $B$ we apply a uniform sampling strategy across all clusters to avoid the sampling process being dominated by large or high-frequency semantic clusters. Within each cluster $C_k$, we adopt a k-center greedy algorithm (Sener & Savarese, 2018) to select a subset $G_k \subset C_k$ of size $M = \lfloor \frac{B}{K} \rfloor$, aiming to minimize the maximum distance from any sample in the cluster to the selected set:

$$\min_{G_k} \max_{v \in C_k} \min_{g \in G_k} \|v - g\|_2 \qquad (2)$$

where $\|\cdot\|_2$ denotes the Euclidean norm. This objective favors retaining samples located in sparse regions or near cluster boundaries while suppressing redundant points in

dense cluster centers, thereby maximizing local geometric diversity.

### 3.3. Stage II: Diagnostic Triage

Stage II performs fine-grained quality assessment and triage of instruction samples. Each sample is assigned to one of three quality states—*High Quality*, *Fixable*, or *Useless*—corresponding to well-formed samples, samples with localized defects, and severely flawed samples.

Stage II consists of two steps: diagnosis and triage. Given an instruction sample $x$, a teacher model $M$ produces a quantitative diagnostic score and a qualitative description:

$$M(x) \rightarrow (s(x), H(x)) \qquad (3)$$

Here, $s(x) = (s_{\text{inst}}, s_{\text{rich}}, s_{\text{sound}}, s_{\text{pres}})$ is a four-dimensional quality diagnostic score, capturing instruction adherence, content richness, logical soundness, and presentation quality (Ye et al., 2024). $H(x)$ is a natural-language diagnostic description of the sample's primary quality defect. These diagnostic outcomes are used in Stage III to guide targeted remediation.

Next, a deterministic triage function assigns each sample to a mutually exclusive category $c$:

$$f_{\text{triage}} : x \rightarrow c, \quad c \in \{High\ Quality, Fixable, Useless\}, \qquad (4)$$

where the assignment is determined by the diagnostic score $s(x)$. The triage rules are defined as follows. *High Quality*

samples meet stringent criteria across all quality dimensions and are therefore retained as-is. *Useless* samples fall below the minimum acceptable threshold on at least one critical quality dimension, indicating deviations that lie beyond the scope where targeted remediation would be effective; these samples are therefore discarded. *Fixable* samples satisfy basic requirements in instruction adherence and logical soundness but exhibit localized defects in one or more specific quality dimensions. Given their retained semantic value, these samples are passed to Stage III for diagnostic-guided targeted remediation. Illustrative examples of samples from each triage category, together with their diagnostic scores and descriptions, are provided in Appendix D. Separately, the full diagnostic prompt, including the scoring rubric and templates, is provided in Appendix A.1.

### 3.4. Stage III: Diagnostic-Guided Targeted Remediation

In Stage III, we apply the teacher model $M$ to perform diagnostic-guided targeted remediation on samples identified as fixable in Stage II. Any sample that falls below the high-quality threshold along one or more quality dimensions is considered to contain a quality deficiency. To mitigate potential semantic drift, we adopt a diagnostic-guided remediation strategy, where modifications are constrained according to each sample's diagnostic outcomes. To maintain the semantic integrity of the instructions, only the response content of each sample is subject to modification.

For each *Fixable* sample $x$, we combine its diagnostic score $s(x)$ and diagnostic description $H(x)$ to construct a remediation guidance:

$$\mathcal{P} = \mathcal{T}(s(x), H(x)), \tag{5}$$

where $H(x)$ summarizes the primary quality deficiencies, and the dimensions of $s(x)$ determine the specific remediation actions. Deficiencies along the quality dimensions are grouped into two types—correctness and expression quality—leading to a two-phase remediation procedure:

**Phase 1: Correctness Restoration.** Applied when a sample exhibits deficiencies in logical soundness $s_{\text{sound}}$ or instruction adherence $s_{\text{inst}}$, this phase corrects factual errors, reasoning flaws, or constraint violations, ensuring that the sample meets basic correctness requirements.

**Phase 2: Content Enrichment and Expression Refinement.** Applied when a sample exhibits deficiencies in content richness $s_{\text{rich}}$ or presentation quality $s_{\text{pres}}$, this phase enhances the depth, clarity, and presentation of the content. All Phase 2 modifications are performed under the constraint that correctness established in Phase 1 is preserved.

Finally, $H(x)$ together with the remediation actions determined by $s(x)$ will construct the remediation guidance $\mathcal{P}$. During remediation, $P$ is incorporated into the prompt for

teacher model $M$, conditioning its generation to produce the remediated sample $x_{\text{Remediated}}$. Detailed implementation specifics and the remediation prompt templates are provided in Appendix A.2. The final instruction dataset is formed by combining all *High Quality* samples with all remediated samples:

$$D_{\text{final}} = D_{\text{High Quality}} \cup D_{\text{Remediated}}. \tag{6}$$

## 4. Experiments

### 4.1. Datasets

**Training Datasets** We conduct experiments on two commonly used instruction-tuning datasets: Alpaca (Taori et al., 2023) and WizardLM (Xu et al., 2024). The Alpaca dataset is constructed using the Self-Instruct framework, with instructions automatically generated from the Davinci-003 model. It is of moderate overall quality and contains 52,002 instruction samples. The WizardLM dataset is built using the Evol-Instruct approach, in which instructions and responses are iteratively evolved using ChatGPT. It is of relatively high overall quality. We adopt existing methodologies to filter samples subject to "AI censure" in WizardLM, resulting in a WizardLM dataset comprising 63,780 instances.

**Evaluation Datasets** We adopt five widely used benchmarks—Vicuna (Chiang et al., 2023), Koala (Vu et al., 2023), WizardLM (Xu et al., 2024), Self-Instruct (Wang et al., 2023), and LIMA (Zhou et al., 2023)—to comprehensively assess instruction-following ability and generalization across diverse tasks. Together, these benchmarks comprise 1,030 carefully curated instruction samples, spanning a wide range of tasks from general question answering to complex reasoning.

### 4.2. Experimental Setting

**Baselines** We compare the proposed Cure-SFT with the following baseline methods: 1) FULL: Instruction fine-tuning using the full original training dataset. 2) Random: Instruction fine-tuning on a randomly sampled subset of the full training data. 3) IFD (Li et al., 2024b): Data selection based on instruction-following difficulty, favoring samples with higher IFD scores. 4) AlpaGasus (Chen et al., 2024b): A method that uses ChatGPT to score the quality of training samples and selects data accordingly. 5) Deita (Liu et al., 2024b): A method that first trains a dedicated LLM-based scorer on ChatGPT-annotated data to assess sample quality and complexity, and then performs data selection based on quality, complexity, and diversity. 6) T-SHIRT (Fu et al., 2026): A method that is based on token-selective hierarchical data selection. 7) Rewrite: A baseline that rewrites responses of randomly sampled samples using a teacher model under a fixed data budget. 8) CoachLM (Liu et al., 2024c): A method that trains a dedicated model on expert

*Table 1.* Performance comparison of different data curation strategies. Higher scores indicate better performance. Best results are shown in **bold**.

| Dataset | Method | Winning Score (vs. FULL) | | | | | | AlpacaEval 2.0 | |
|---------|--------|----------|--------|-------|------|-----------|------|--------|-----|
| | | WizardLM | Vicuna | Koala | LIMA | Sinstruct | Avg. | LC Win | Win |
| Alpaca (10%) | Random | 1.11 | 1.18 | 0.96 | 0.97 | 0.88 | 1.02 | 9.75 | 3.65 |
| | IFD | 1.24 | 1.36 | 1.37 | 1.28 | 1.03 | 1.26 | 6.21 | 2.92 |
| | Deita | 1.10 | 1.29 | 1.26 | 1.24 | 0.94 | 1.17 | 7.82 | 3.14 |
| | Alpagasus | 1.09 | 1.09 | 1.11 | 1.14 | 0.91 | 1.07 | 9.83 | 3.84 |
| | T-SHIRT | 1.01 | 1.36 | 1.09 | 1.01 | 0.86 | 1.07 | 9.90 | 3.95 |
| | Rewrite | 1.62 | 1.74 | 1.48 | 1.69 | 1.17 | 1.54 | **12.98** | 6.68 |
| | CoachLM | 1.17 | 1.20 | 1.12 | 1.05 | 0.99 | 1.11 | 10.32 | 4.23 |
| | Selective-RT | 1.63 | 1.85 | 1.52 | 1.74 | 1.39 | 1.63 | 4.08 | 3.48 |
| | NILE | 1.64 | 1.90 | 1.66 | 1.79 | 1.46 | 1.69 | 11.85 | 7.31 |
| | **Cure-SFT** | **1.85** | **1.91** | **1.78** | **1.83** | **1.52** | **1.78** | 11.45 | **7.66** |
| WizardLM (10%) | Random | 0.91 | 0.84 | 0.87 | 0.95 | 0.73 | 0.86 | 8.10 | 3.77 |
| | IFD | 0.65 | 0.46 | 0.76 | 0.60 | 0.88 | 0.67 | 5.78 | 3.04 |
| | Deita | 0.64 | 0.33 | 0.81 | 0.54 | 0.79 | 0.62 | 7.27 | 3.45 |
| | Alpagasus | 0.86 | 0.86 | 0.97 | 0.92 | 0.83 | 0.89 | 7.39 | 4.03 |
| | T-SHIRT | 0.82 | 1.01 | 0.89 | 0.84 | 0.82 | 0.88 | 9.15 | 4.13 |
| | Rewrite | 0.93 | 1.25 | 1.15 | **1.34** | 0.93 | 1.12 | **10.04** | 5.70 |
| | CoachLM | 0.84 | 1.01 | 0.97 | 0.96 | 0.87 | 0.93 | 7.84 | 4.76 |
| | Selective-RT | 1.15 | 1.25 | 1.22 | 1.32 | 1.16 | 1.22 | 4.81 | 4.37 |
| | NILE | 1.24 | **1.29** | 1.27 | 1.22 | 1.13 | 1.23 | 9.45 | 7.14 |
| | **Cure-SFT** | **1.32** | 1.26 | **1.34** | 1.26 | **1.19** | **1.27** | 9.67 | **7.24** |

revisions for automatic instruction correction. 9) Selective-RT (Li et al., 2024a): A method that combines reflection-based rewriting with student-side selection. 10) NILE (Hu et al., 2025): A method that revises responses to better align with the model's internal knowledge.

For Rewrite, we use the same teacher model as in Cure-SFT, and the rewriting prompts follow the Alpaca-GPT4 format. Except for FULL, all methods are evaluated under the same data budget to ensure a fair comparison.

**Implementation Details of Cure-SFT** We use the pre-trained BGE-M3 (Chen et al., 2024a) model as the encoder for stratified semantic-geometric sampling, and a locally deployed Qwen2.5-72B-Instruct-AWQ model as the teacher. For the diagnostic score, each quality dimension is rated on a discrete integer scale from 1 to 5. Based on the resulting diagnostic score, we adopt a deterministic triage strategy with fixed thresholds: A sample is classified as *High Quality* if it satisfies $s_{inst} = 5 \land s_{rich} \geq 4 \land s_{sound} \geq 4 \land s_{pres} \geq 3$; A sample is categorized as *Useless* if $s_{inst} < 4 \lor s_{rich} < 3 \lor s_{sound} < 3$. All remaining samples are treated as *Fixable*. These thresholds follow the scoring rubric; further details are in Appendix B. The threshold sensitivity experiment is provided in Appendix E.1. While baseline methods strictly use the full data budget, Cure-SFT results in a slightly smaller final training set because samples classified as *Useless* are discarded without replacement.

**Instruction Tuning Details** We adopt LLaMA-3-8B as the base model for instruction fine-tuning. Training is conducted using the LoRA (Hu et al., 2022) approach, with the LoRA rank set to 16. All experiments are trained for three epochs, using a learning rate of 2e-4 and a global batch size of 64. We also conduct experiments using Mistral-7B as the base model; results are reported in Appendix E.2.

### 4.3. Evaluation and Benchmark

**Pairwise Evaluation** In open-ended instruction generation tasks, model outputs exhibit substantial diversity, making automatic evaluation based on fixed reference answers inadequate for fully capturing instruction-following ability and generation quality. Following recent work on LLM-as-a-Judge (Zheng et al., 2023), we use a stronger large language model as a judge to conduct pairwise comparisons of responses generated by different models. We adopt the overall winning score as the evaluation metric, defined as: $(Num(Win) - Num(Lose))/Num(All) + 1$. The detailed evaluation protocol is provided in Appendix C.

**Benchmark** We also conduct standardized evaluation using the widely adopted AlpacaEval 2.0 (Dubois et al., 2024) benchmark. This benchmark is built on the Alpaca-Farm (Dubois et al., 2023) dataset and uses GPT-4-Turbo as an automated evaluator to measure a model's win rate and length-controlled win rate (LC Win Rate) relative to GPT-4.

*Table 2.* Component-wise ablation study on the Alpaca dataset. We analyze the contribution of each module in Cure-SFT. A checkmark (✔) indicates the component is enabled, while a cross (✘) indicates it is removed. The best results are highlighted in **bold**.

| Components | | | Winning Score (vs. FULL) | | | | | |
|---|---|---|---|---|---|---|---|---|
| **Stage I** | **Stage II** | **Stage III** | WizardLM | Vicuna | Koala | LIMA | Sinstruct | Avg. |
| ✘ | ✔ | ✔ | 1.73 | **1.93** | 1.70 | 1.76 | 1.51 | 1.72 |
| ✔ | ✘ | ✘ | 0.93 | 1.34 | 1.06 | 1.06 | 0.86 | 1.05 |
| ✔ | ✔ | ✘ | 1.13 | 1.36 | 1.16 | 1.24 | 1.08 | 1.19 |
| ✔ | ✔ | ✔ | **1.85** | 1.91 | **1.78** | **1.83** | **1.52** | **1.78** |

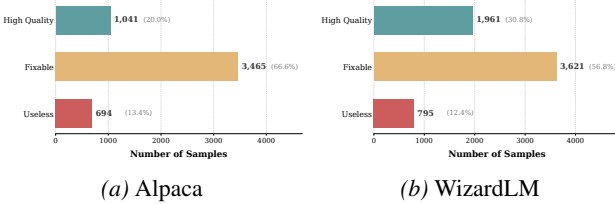

*(a)* Alpaca       *(b)* WizardLM

*Figure 2.* Distribution of diagnostic triage results on (a) Alpaca and (b) WizardLM datasets under a 10% data budget.

## 4.4. Main Results

Table 1 presents a performance comparison of different instruction data curation methods under a 10% data budget.

**Cure-SFT consistently outperforms baselines.** The results show that Cure-SFT achieves the best or highly competitive performance across the majority of evaluation metrics on both the Alpaca and WizardLM datasets. In contrast, several selection-based methods, such as IFD and Deita, exhibit unstable behavior across datasets. While they often outperform random sampling on Alpaca, their average Winning Score on the higher-quality WizardLM dataset falls below that of random sampling. These results indicate that purely selection-based strategies can be sensitive to the underlying data distribution and may underperform when applied to more complex or higher-quality instruction data.

**Targeted Remediation vs. Unified Rewriting** When compared with the Rewrite baseline, Cure-SFT exhibits more stable and consistent gains on the fine-grained Winning Score across multiple benchmarks, even though Rewrite achieves strong LC Win Rate on AlpacaEval 2.0. This contrast underscores a difference in performance stability. Although unified rewriting can achieve strong results on specific benchmarks, its gains are less consistent across evaluation settings. In contrast, Cure-SFT delivers more stable improvements on fine-grained Winning Scores across diverse benchmarks.

**Robustness to Varying Initial Data Quality** Cure-SFT demonstrates consistent performance gains across datasets with different initial quality levels. The improvements are more pronounced on Alpaca, which contains a larger propor-

*Table 3.* Comparison of Winning Scores across different selection scales.

| Metric | Winning Score (vs. FULL) | | | |
|---|---|---|---|---|
| **Selection Scale** | **1%** | **5%** | **10%** | **15%** |
| Random | 0.68 | 1.01 | 1.11 | 1.01 |
| IFD Score | 1.06 | 1.27 | 1.24 | 1.27 |
| Alpagasus | 0.95 | 1.04 | 1.09 | 1.13 |
| Deita | 1.13 | 1.17 | 1.10 | 1.00 |
| T-SHIRT | 0.99 | 1.08 | 1.01 | 1.04 |
| Rewrite | 1.41 | 1.58 | 1.62 | 1.60 |
| CoachLM | 0.70 | 0.88 | 1.17 | 1.21 |
| Selective-RT | 1.44 | 1.59 | 1.63 | 1.66 |
| NILE | 1.47 | 1.66 | 1.64 | 1.69 |
| **Cure-SFT** | **1.60** | **1.70** | **1.85** | **1.78** |

tion of imperfect samples, while still remaining positive on the higher-quality WizardLM dataset. This pattern suggests that Cure-SFT effectively exploits remediation opportunities in noisier data regimes without degrading performance when applied to already high-quality instruction data.

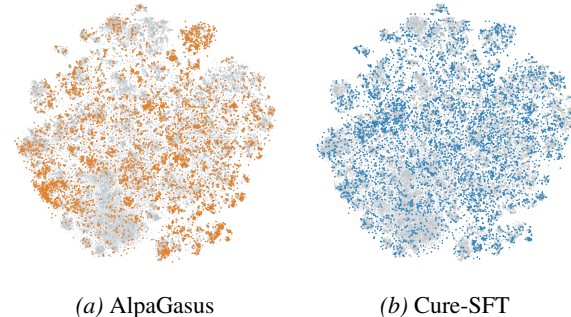

*(a)* AlpaGasus       *(b)* Cure-SFT

*Figure 3.* Visualization of semantic diversity on Alpaca dataset under a 10% data budget.

## 4.5. Ablation Study

To better understand the contribution of each stage in Cure-SFT to performance improvements, we conduct component-level ablation studies on the Alpaca dataset. The results are reported in Table 2. Furthermore, a controlled ablation

on the *Fixable* subset comparing targeted remediation with unguided rewriting is provided in Appendix E.3.

**Effect of Removing Stratified Semantic-Geometric Sampling** In this variant, we replace Stratified Semantic-Geometric Sampling with random sampling, while still applying quality diagnosis and targeted remediation within the randomly selected subset. This setting remains competitive across multiple benchmarks, and even yields slightly better results on benchmarks such as Vicuna, which are biased toward more common instruction patterns. However, on benchmarks such as WizardLM and LIMA that involve complex reasoning or long-tail instructions, the full Cure-SFT pipeline consistently maintains an advantage. This observation suggests that, without explicit semantic stratification constraints, even when combined with quality diagnosis and remediation, the resulting training data may exhibit imbalanced coverage in the semantic space, which in turn can limit the model's generalization to complex or rare instructions.

**Effect of Removing Diagnostic Triage and Targeted Remediation** Since the targeted remediation in Stage III depends on the diagnostic description produced by Stage II, removing Stage II also entails removing Stage III, leaving only Stage I in this variant. A further comparison between the variants w/o Stage II & III and w/o Stage III shows that model performance continues to decline, with the average Winning Score dropping to 1.05. This gap indicates that, while Stratified Semantic-Geometric Sampling helps maintain diversity in the instruction distribution, in the absence of a quality diagnosis mechanism, a substantial number of low-quality samples may still be included in the training data, which can substantially undermine model performance. These results suggest that Stage II plays a foundational role in quality control within the overall pipeline, providing a reliable basis for subsequent data remediation.

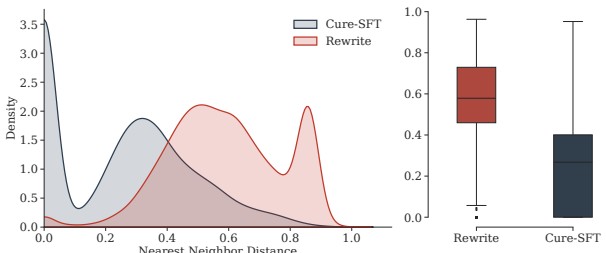

*Figure 4.* Distance distributions between processed samples and their nearest original neighbors in feature space on WizardLM dataset under a 10% data budget for Cure-SFT and Rewrite.

**Effect of Removing Diagnostic-Guided Targeted Remediation** This variant retains Stage I and Stage II while removing diagnostic-guided targeted remediation, meaning that only *High Quality* samples are used for training. This result suggests that relying solely on selection without remediation

can, to some extent, limit both the effective size of the training data and its utilization. Some samples, while exhibiting localized deficiencies in the overall quality assessment, still contain information that is valuable for instruction alignment. Through diagnostic-guided targeted remediation, the full method improves the usability of such samples, thereby further enhancing the overall effectiveness of the training data.

## 4.6. Analysis

### 4.6.1. DISTRIBUTION OF DIAGNOSTIC TRIAGE

To analyze the diagnostic behavior of Cure-SFT, we examine the distribution of *High Quality*, *Fixable*, and *Useless* samples in the Alpaca and WizardLM datasets under a 10% data budget, as shown in Figure 2. Overall, the diagnostic triage of Cure-SFT primarily favors *Fixable* samples, while also retaining high-quality data and selectively filtering out low-value samples. Across both datasets, *Fixable* samples account for the largest proportion, suggesting that raw instruction data often remains semantically valid overall while exhibiting localized deficiencies. By contrast, WizardLM exhibits a higher proportion of *High Quality* samples, reflecting its superior overall data quality and a reduced reliance on remediation. Meanwhile, the proportion of *Useless* samples remains low and comparable across both datasets, indicating that Cure-SFT can consistently identify and filter out low-value or hard-to-remediate data. Taken together, these distributional patterns indicate that the diagnostic triage of Cure-SFT adapts to variations in data quality.

### 4.6.2. ANALYSIS OF PERFORMANCE UNDER DIFFERENT DATA BUDGETS

As shown in Table 3, we consider multiple data budgets on the Alpaca dataset and evaluate the Winning Score on the WizardLM test set, using full-data fine-tuning as the reference, to examine how changes in data budget affect model performance. Under extremely low-resource settings, Cure-SFT still delivers substantial gains, reaching a Winning Score of 1.60 with only 1% of the training data, which is on par with the performance of Rewrite at a 15% data budget. As the data budget increases, the performance of Cure-SFT continues to improve steadily and begins to saturate around the 10% budget, a pattern that suggests high-value fixable signals are already exploited early on. When the budget is further expanded, newly included samples exhibit increasingly similar remediation patterns, yielding limited marginal information gain, which in turn slows further performance improvements. Nevertheless, Cure-SFT consistently outperforms competing methods across all budget levels and exhibits more robust overall performance.

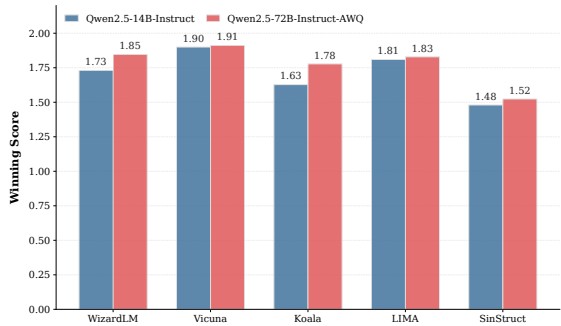

*Figure 5.* Cure-SFT performance on Alpaca dataset under a 10% data budget across different teacher model sizes.

### 4.6.3. ANALYSIS OF DIVERSITY AND DISTRIBUTION INTEGRITY

To analyze how different data curation strategies affect the structure of the semantic representation space, we conduct both qualitative and quantitative analyses.

**Visualizing Semantic Diversity** As shown in Figure 3, we visualize samples from the Alpaca dataset under a 10% data budget in a two-dimensional semantic space using t-SNE. The coverage of Cure-SFT and AlpaGasus is compared, with the full original dataset distribution shown as the background. AlpaGasus produces relatively compact clusters in the embedding space, with noticeable gaps in certain regions, indicating a tendency to favor specific types of samples and, as a result, to diminish semantic diversity to some extent. In contrast, Cure-SFT exhibits a more dispersed sample distribution, whose coverage closely aligns with the original data distribution, suggesting better preservation of the semantic space structure during the sampling stage.

**Quantifying Distribution Fidelity** To quantify the semantic impact of the rewriting process, we compute the Euclidean distance in feature space between each processed sample and its nearest original neighbor on WizardLM dataset under a 10% data budget. Figure 4 shows the resulting distance distributions. Cure-SFT exhibits a pronounced density near zero distance, with additional mass concentrated in small-distance regions, indicating that it preserves a substantial portion of high-quality original samples while applying limited, targeted corrections to others. This distribution reflects a favorable trade-off between quality improvement and distribution fidelity. In contrast, the Rewrite baseline shows a systematic shift toward larger distances, suggesting that uniform rewriting induces greater perturbations to the original semantic structure.

### 4.6.4. EFFECT OF TEACHER MODEL SIZE

To examine the robustness and generalization of Cure-SFT, we evaluate its performance with teacher models of differ-

---

**Instruction**

Output a Java code segment that takes an array of numbers and returns the median value.

**Original Response**

```java
public static double getMedian(int[] a)
{
    Arrays.sort(a);
    int len = a.length;
    if (len % 2 == 0)
        return (a[(len/2)] + a[(len/2)-1]) / 2;
    else
        return a[len/2];
}
```

Looks correct at first glance, but there's a subtle issue.

**Diagnosis Description**

The response does not handle edge cases, such as an empty array or non-integer values.
The division for the median calculation should be a floating-point division to ensure accuracy.

**Remediated Response**

```java
public static double getMedian(int[] a) {
    if (a == null || a.length == 0) {
        throw new IllegalArgumentException("Array must not be null or empty");
    }
    Arrays.sort(a);
    int len = a.length;
    if (len % 2 == 0) {
        return (a[len / 2] + a[(len / 2) - 1]) / 2.0;
    } else {
        return a[len / 2];
    }
}
```

Targeted Remediation

*Figure 6.* A case study of Cure-SFT applying diagnostic-guided targeted remediations to a Java code generation example.

ent sizes on the Alpaca dataset under a 10% data budget. Figure 5 reports the winning scores obtained using Qwen-2.5-14B-Instruct and Qwen-2.5-72B-Instruct-AWQ. With the smaller 14B teacher, Cure-SFT consistently improves instruction data quality, achieving winning scores between 1.48 and 1.90 across all evaluation splits. While the larger 72B teacher yields slightly higher scores, the performance gap is relatively small, indicating that Cure-SFT's diagnostic triage and targeted remediation generalize well across teacher capacities. Overall, Cure-SFT delivers stable gains with both high-capacity and weaker teachers, demonstrating robustness and practical applicability in settings where access to very large models is limited.

### 4.7. Case Study

To provide a more concrete illustration of Cure-SFT's remediation behavior on a real instruction instance, we present a specific example in Figure 6. The instruction asks for generating a piece of Java code to compute the median of an array. The original response is logically sound at a high level but contains two subtle semantic errors, related to boundary condition handling and numerical precision, respectively. Cure-SFT first identifies the specific defects present in a given sample and assesses whether the sample is amenable to remediation. It then performs targeted remediation for these defects, thereby correcting potential errors without altering the original code structure or the underlying implementation logic. This example illustrates that

Cure-SFT's diagnostic-guided targeted remediation mechanism can effectively improve sample quality, while keeping remediation-induced semantic drift under control to a reasonable extent.

## 5. Conclusion

This paper studies the problem of data curation for instruction tuning and proposes Cure-SFT, a diagnostic-guided instruction data curation method. Cure-SFT follows a coarse-to-fine curation paradigm that explicitly distinguishes different sample quality states within a unified pipeline. Rather than treating all imperfect samples uniformly, it applies diagnostic-guided targeted remediation to samples that retain core semantic value, while filtering out low-utility instances. Extensive experiments on public instruction datasets, including Alpaca and WizardLM, across multiple evaluation benchmarks demonstrate that Cure-SFT achieves consistent performance gains under varying data budget settings. More broadly, Cure-SFT advocates a data curation philosophy that emphasizes repairing samples instead of simply discarding or rewriting them. This perspective enables a more fine-grained and controllable approach to instruction tuning, allowing quality improvements while preserving data diversity and distributional fidelity.

## Impact Statement

This paper presents research aimed at advancing instruction tuning for large language models. The proposed method focuses on diagnostic-guided data curation, enabling more efficient use of existing instruction data by identifying, repairing, or filtering samples according to their quality characteristics. As with most advances in large language model training, the techniques introduced in this work may have broader societal implications depending on how the resulting models are deployed. However, the methods proposed here operate at the level of data curation and optimization and do not introduce new model capabilities or application domains beyond those already widely studied. We therefore believe that there are no immediate ethical concerns specific to this work that require detailed discussion beyond existing considerations in large-scale language model training.

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

# A. Methodological Supplement

## A.1. Diagnostic Prompt Template for Stage II

*Figure 7.* The diagnostic prompt template used for Stage II.

---

**Diagnostic Triage Prompt**

---

You are an expert Data Quality Auditor.
Your task is to DIAGNOSE quality defects in a model response.
You do NOT rewrite, fix, or optimize the response.
Your feedback is diagnostic and NON-BINDING, used by an automated refinement system.

**Task**
Given an (Instruction, Input, Response) triple:
1. Assign quality scores across four independent dimensions.
2. Identify the PRIMARY limiting factor that prevents the response from being near-perfect.
3. Provide a non-binding description describing what information appears to be missing or weak.

**Scoring Principles**
- Each dimension is evaluated independently.
- Use the full 1-5 scale (5 = near-perfect). Avoid defaulting to 3.
- Score meanings: 2 = substandard effort; 3 = passable but limited.

**Scoring Rubric**
*1. Instruction Compliance*
- 5: Fully follows instruction and all constraints.
- 4: Minor looseness in interpreting vague constraints.
- 3: Misses a minor constraint.
- 2: Violates a hard constraint.
- 1: Fails to address the task.

*2. Information Richness (Focus: Depth, specificity, structure)*
- 5: Insightful, specific, includes reasoning, examples, or limitations.
- 4: Adequate depth, at least one concrete supporting element.
- 3: Correct but generic; surface-level explanation.
- 2: Sparse or underdeveloped.
- 1: Empty or meaningless.

*3. Logical Soundness*
- 5: Premises are factually true AND reasoning is flawless.
- 4: Premises are true, reasoning has minor imprecision.
- 3: Premises are true, but reasoning steps are incomplete.
- 2: The core factual premise is WRONG, even if the reasoning flows logically.
- 1: Complete fabrication or nonsensical content.

*4. Presentation Quality*
- 5: Clear structure, effective formatting.
- 4: Readable but could be better organized.
- 3: Wall-of-text but understandable.
- 2: Formatting harms readability.
- 1: Unreadable.

**Input**
Instruction: {instruction}\nInput Context: {input}\nModel Response: {output}

**Output Format (JSON Only)**
{{
"scores": {{
"compliance": 1-5,
"richness": 1-5,
"soundness": 1-5,
"presentation": 1-5
}},
"diagnostic_description": "A brief, non-binding description of what seems missing or weak. Required unless all scores are 5."
}}

---

## A.2. Diagnostic-Guided Targeted Remediation

### A.2.1. OVERVIEW OF DIAGNOSTIC-GUIDED REMEDIATION

In Stage III, we perform diagnostic-guided targeted remediation on samples identified as fixable in Stage II. Rather than freely rewriting responses, the remediation process is explicitly constrained by diagnostic signals, including a diagnostic score that localizes dimension-specific quality deficiencies and a diagnostic description that summarizes the primary quality bottleneck. These diagnostic results are translated into structured remediation actions with a defined execution priority, which are then encoded as natural-language guidance to condition the teacher model's generation, enabling targeted correction while preserving the original instruction semantics.

### A.2.2. DIMENSION-LEVEL REMEDIATION ACTIONS

Each dimension of the diagnostic score $s(x)$ is associated with a corresponding category of remediation action when its value falls below the predefined quality criterion. Table 4 summarizes the mapping from dimension-level quality deficiencies to remediation actions. When multiple quality deficiencies are identified across different dimensions, the corresponding remediation actions are jointly reflected in the constructed remediation guidance.

### A.2.3. TWO-PHASE ORGANIZATION AND PRIORITY STRUCTURE

The remediation actions associated with different quality dimensions are organized according to their optimization objectives within a two-phase remediation framework. Remediation behaviors related to correctness and constraint satisfaction, including soundness and instruction compliance, are grouped into *Phase 1 (Correctness Restoration)*. Remediation behaviors focusing on information adequacy and expression quality, including richness and presentation, are grouped into *Phase 2 (Content Enrichment and Expression Refinement)*. The two-phase organization follows a correctness-first principle: all refinements associated with Phase 2 are designed to preserve and remain consistent with the correctness established in Phase 1.

### A.2.4. REMEDIATION GUIDANCE

Building upon the dimension-level remediation actions and their two-phase organization, Stage III encodes the diagnostic outcomes into an explicit remediation guidance. For each *Fixable* sample $x$, Stage III constructs an explicit remediation guidance that specifies how the response should be revised based on the diagnostic outcomes obtained in Stage II.

For each *Fixable* sample $x$, we combine its diagnostic score $s(x)$ and diagnostic description $H(x)$ to construct a remediation guidance:

$$\mathcal{P} = \mathcal{T}(s(x), H(x)), \tag{7}$$

where $H(x)$ summarizes the primary quality deficiencies, and the dimensions of $s(x)$ determine the specific remediation actions.

### A.2.5. CONSTRUCTION OF REMEDIATION GUIDANCE

The diagnostic description $H(x)$, together with the remediation behaviors implied by the diagnostic score $s(x)$ and global remediation constraints, are integrated into a structured, natural-language remediation guidance $\mathcal{P}$. Figure 8 presents an illustrative example of the resulting remediation guidance.

### A.2.6. REMEDIATION PROMPT

The remediation guidance $\mathcal{P}$ is incorporated as an explicit guidance segment within the remediation prompt. Along with the original instruction, input context, and the response to be remediated, the prompt is fed to the teacher model $M$ in a single inference pass, where $\mathcal{P}$ serves to condition the generation process. The complete remediation prompt template is provided in Figure 9.

*Table 4.* Mapping between diagnostic score and corresponding remediation actions.

| Quality Dimension | Condition | Remediation Action |
|---|---|---|
| Instruction Compliance | $s_{inst} < 5$ | Enforce user constraints (format, length, negatives). Rewrite affected sections to strictly satisfy them. |
| Logic Soundness | $s_{sound} < 4$ | Scrutinize premises. If a premise is FALSE, correct it immediately. If reasoning relies on weak assumptions, make them explicit or rigorously derived. |
| Information Richness | $s_{rich} < 4$ | Vertically expand the explanation. Instantiate abstract claims with concrete examples. |
| Presentation Quality | $s_{pres} < 3$ | Reorganize text into clear hierarchy (headers, bullets). |

*Figure 8.* Example of remediation guidance generated from diagnostic results.

**Remediation Guidance**

**Diagnostic Description:**
{*diagnostic_description*}

**Phase 1: Correctness Restoration**
(Focus: Correctness & Compliance. NO Fluff.)
1. [LOGIC/FACT] Scrutinize premises. If a premise is FALSE, correct it immediately. If reasoning relies on weak assumptions, make them explicit or rigorously derived.
2. [ALIGNMENT] Enforce user constraints (format, length, negatives). Rewrite affected sections to strictly satisfy them.

**Phase 2: Content Enrichment and Expression Refinement**
(Focus: Depth & Readability. Only proceed if Phase 1 is secure.)
1. [DEPTH] Vertically expand the explanation. Instantiate abstract claims with concrete examples.
2. [FORMAT] Reorganize text into clear hierarchy (headers, bullets).

**Global Constraints**
– FACTUAL SUPREMACY: Accuracy overrides all other instructions.
– Minimal Intervention: Do not rewrite parts that are already perfect.
– Integration: Phase 2 edits must be woven seamlessly into the Phase 1 base.

*Figure 9.* Remediation prompt used for Stage III.

---

**Remediation Prompt**

---

You are a precision editor for LLM-generated data.
You do not freely rewrite text.
You revise the response strictly following the remediation guidance below, using a two-phase process.

**Input Data**
User Instruction: {instruction}
User Context: {input}
Original Response: {old_output}

**Remediation Guidance**
{remediation_guidance}

**Execution**
Step 1: *Repair Plan*
– Identify which remediation actions are applied in Phase 1 and Phase 2.
– Explicitly state how you will resolve conflicts (e.g., Accuracy > Alignment).
Step 2: *Refined Response*
– Produce the final remediated output enclosed within <remediated_response> tags.
– Apply Phase 2 modifications only after Phase 1 corrections are completed.

**Output Format**
Repair Plan:
<brief plan>
<remediated_response>
[Final refined response]
</remediated_response>

---

## B. Design Rationale for Deterministic Triage Thresholds

This section provides additional clarification on the design rationale of the deterministic triage thresholds adopted in Stage II. Specifically, we describe how these thresholds are derived from the scoring rubric used for diagnostic evaluation and how they relate to the semantic interpretation of the resulting diagnostic scores. The scoring rubric itself is explicitly embedded in the diagnostic prompt provided to the teacher model.

### B.1. Scoring Rubric and Discrete Semantic States

Figure 7 presents the full diagnostic prompt, including the rubric definitions for each score level across all evaluation dimensions. Each dimension is defined using discrete, semantically grounded criteria associated with integer scores from 1 to 5, rather than a continuous notion of quality. Under this formulation, the triage thresholds are follow naturally from the discrete semantic distinctions specified in the scoring rubric. Each threshold corresponds to a boundary between qualitatively different semantic states as defined in the diagnostic prompt.

### B.2. Thresholds as Semantic Constraints for Triage

Under the above interpretation, the triage thresholds in Stage II act as semantic constraints that separate qualitatively different categories of samples. Each threshold reflects a minimum semantic requirement associated with the corresponding rubric definition.

The *Useless* category is determined by a set of exclusionary conditions. Specifically, a sample is classified as *Useless* if it violates the minimum semantic requirements on any critical dimension, namely $s_{\text{inst}} < 4$, $s_{\text{rich}} < 3$, or $s_{\text{sound}} < 3$. According to the rubric, such scores correspond to fundamental issues such as violations of instruction constraints, insufficient semantic content, or incorrect or incoherent reasoning. These deficiencies compromise the core usability of the sample and therefore preclude effective remediation.

At the other end of the spectrum, *High Quality* samples are identified using a stricter conjunction of semantic requirements. The condition $s_{\text{inst}} = 5$ enforces complete and unambiguous adherence to all instruction constraints, as explicitly defined by the rubric. Additional thresholds on logical soundness, information richness, and presentation quality ensure that samples in this category are not only correct, but also sufficiently informative and clearly presented. Samples satisfying all these criteria are retained without modification.

All remaining samples that satisfy the exclusion constraints but do not meet the full *High Quality* criteria are classified as *Fixable*. By construction, these samples preserve core semantic validity and task intent, while exhibiting localized and well-characterized deficiencies. This intermediate category corresponds to samples for which diagnostic-guided targeted remediation is feasible and beneficial, and such samples are therefore passed to Stage III.

### B.3. Component-wise Triage versus Aggregate Scoring

The use of component-wise thresholds follows naturally from the rubric design. Each evaluation dimension captures a distinct semantic property, and aggregating scores across dimensions can obscure critical deficiencies. For example, high presentation quality or information richness cannot compensate for violations in logical soundness or instruction compliance.

Component-wise triage ensures that each retained or remediated sample satisfies minimum semantic requirements along all critical dimensions. This design preserves the interpretability of the triage process and aligns sample categorization with their diagnostic profiles.

# C. Evaluation Details

For each instruction in the test set, we generate responses from model A and model B, which are then evaluated by a judge model on overall quality. The evaluation considers multiple dimensions, including helpfulness, accuracy, relevance, level of detail, and safety. Each response is assigned an overall score on a 1–10 scale, where higher scores indicate better performance.

Following the LLM-as-a-Judge paradigm, we employ GPT-4.1-mini as the judge model. This model exhibits evaluation performance comparable to GPT-4o, while offering significantly lower inference cost. The evaluation prompt provided to the judge model follows a fixed template, as illustrated in Figure 10.

To mitigate potential positional bias, each response pair is evaluated twice with the presentation order swapped. The final judgment for each instruction is determined by aggregating the two evaluation results according to the following rules:

**Wins**: The model scores higher in both comparisons, or wins once and ties once.
**Ties**: The model ties in both comparisons, or wins once and loses once.
**Loses**: The model scores lower in both comparisons, or ties once and loses once.

Finally, we adopt the overall winning score as the evaluation metric, defined as:

$$\text{Winning Score} = \frac{N_{\text{Win}} - N_{\text{Lose}}}{N_{\text{All}}} + 1 \tag{8}$$

*Figure 10.* The prompt template used for pairwise evaluation of the model response quality.

---

**Pairwise Evaluation Prompt for Response**

---

**System Prompt**
You are a helpful and precise assistant for checking the quality of the answer.

**User Prompt**
[Question]
{instruction}
[The Start of Assistant 1's Answer]
{response 1}
[The End of Assistant 1's Answer]
[The Start of Assistant 2's Answer]
{response 2}
[The End of Assistant 2's Answer]

We would like to request your feedback on the performance of two AI assistants in response to the user question displayed above.
Please rate the helpfulness, relevance, accuracy, level of details of their responses. Each assistant receives an overall score on a scale of 1 to 10, where a higher score indicates better overall performance. Please first output a single line containing only two values indicating the scores for Assistant 1 and 2, respectively. The two scores are separated by a space. In the subsequent line, please provide a comprehensive explanation of your evaluation, avoiding any potential bias and ensuring that the order in which the responses were presented does not affect your judgment.

---

# D. Illustrative Examples of Diagnostic Outcomes

We include more examples diagnosed by Cure-SFT in this section. The examples of *High Quality*, *Fixable*, *Useless* are shown in Figure 11, Figure 12, and Figure 13, respectively.

---

*Figure 11.* The *High Quality* examples in Alpaca dataset.

---

[Instruction] List four reasons why it would be beneficial to take a gap year.

[Input] None

[Response]

1. Exploration of personal interests. Taking a gap year offers students the opportunity to pursue activities and interests that would not traditionally be taught in the classroom. This can include learning a new language, working on self-development, or engaging in a sport or hobby.

2. Travel. A gap year provides a great opportunity to travel and explore new countries or cultures. This helps young people become more independent, self-aware, and culturally aware.

3. Immersion in a new environment. Taking a gap year allows students the chance to live in a completely new environment, such as a volunteer abroad program, in order to gain life skills and valuable work experience.

4. Clear academic and career choice. Taking a gap year provides a chance for students to think about their desired course of study and career path in a clear and focused way. It allows them to take time away from academic pressure, and instead concentrate on gaining a better understanding of their own values, interests, and strengths.

[Diagnostic Score]

$$s_{\text{inst}} = 5, \ s_{\text{rich}} = 4, \ s_{\text{sound}} = 5, \ s_{\text{pres}} = 4$$

[Diagnostic Description] The response could benefit from more specific examples or data to support the points, such as statistics on the benefits of gap years or real-world success stories.

---

[Instruction] What are some best practices for debugging an AI algorithm?

[Input] None

[Response]

1. Inspect the data set for potential sources of error.

2. Ensure that the data and model are compatible with each other and are pre-processed in the same way.

3. Test each component of the AI system independently.

4. Consider different metrics and data visualizations to better understand the results.

5. Utilize techniques such as bagging or cross-validation to find the source of error.

6. Leverage transfer learning to compare the performance of an AI algorithm to a baseline.

7. Compare the performance of the AI algorithm against different datasets.

8. Use an evolutionary process to quickly try out different combinations to determine the best performing setup.

[Diagnostic Score]

$$s_{\text{inst}} = 5, \ s_{\text{rich}} = 4, \ s_{\text{sound}} = 4, \ s_{\text{pres}} = 4$$

[Diagnostic Description] The response could benefit from more specific examples or detailed explanations for each best practice, particularly for techniques like bagging, cross-validation, and transfer learning.

---

*Figure 12.* The *Fixable* examples in Alpaca dataset.

[Instruction] Rank the following in terms of importance
[Input] Jobs, education, health
[Response]
In terms of importance, health should be ranked first, followed by education, and then jobs.
[Diagnostic Score]
$$s_{\text{inst}} = 5, \ s_{\text{rich}} = 3, \ s_{\text{sound}} = 3, \ s_{\text{pres}} = 5$$

[Diagnostic Description] The response lacks detailed reasoning or evidence to support the ranking of health, education, and jobs. It could benefit from specific examples or data to justify the order.

[Instruction] To what extent is gentrification beneficial to a city?
[Input] None
[Response] Gentrification can bring certain benefits to cities, such as investments in infrastructure, growth in local economy, and an overall improvement in the quality of life. However, gentrification can also have a negative impact, such as a displacement of the existing community, increased housing and cost of living, and a disruption of the culture of the area. Therefore, it is important to thoroughly weigh the pros and cons before jumping to conclusions about the benefits of gentrification for a particular city.
[Diagnostic Score]
$$s_{\text{inst}} = 4, \ s_{\text{rich}} = 3, \ s_{\text{sound}} = 4, \ s_{\text{pres}} = 4$$

[Diagnostic Description] The response could benefit from more specific examples of the benefits and drawbacks of gentrification, as well as a discussion of potential mitigating strategies to address the negative impacts.

*Figure 13.* The *Useless* examples in Alpaca dataset.

[Instruction] Find the difference in temperatures between the following locations.
[Input] Tokyo, Japan and Cairo, Egypt
[Response] The difference in temperature between Tokyo, Japan and Cairo, Egypt is approximately 11 degrees Fahrenheit. Tokyo, Japan has an average temperature of 52.6 degrees Fahrenheit, while Cairo, Egypt has an average temperature of 41.6 degrees Fahrenheit.
[Diagnostic Score]
$$s_{\text{inst}} = 4, \ s_{\text{rich}} = 3, \ s_{\text{sound}} = 2, \ s_{\text{pres}} = 4$$

[Diagnostic Description] The average temperatures provided for Tokyo and Cairo are incorrect. The correct average temperatures are approximately 61.5°F for Tokyo and 77.5°F for Cairo.

[Instruction] Evaluate the statement on a scale of 1 to 5.
[Input] Computer Science is the study of computers and computing.
[Response] 5
[Diagnostic Score]
$$s_{\text{inst}} = 2, \ s_{\text{rich}} = 1, \ s_{\text{sound}} = 2, \ s_{\text{pres}} = 3$$

[Diagnostic Description] The response is a single number without any explanation or context. It fails to address the task of evaluating the statement and lacks any supporting information or reasoning.

# E. ExtraExperiments

## E.1. Sensitivity Analysis of Triage Scoring Thresholds

In the diagnostic triage stage, the thresholds used to categorize samples into High-Quality, Fixable, and Useless states are based on predefined semantic rubrics. To examine the sensitivity of the proposed method to these specific threshold choices, we systematically vary the strictness of the scoring rules around the default configuration.

Specifically, we test looser and stricter conditions for both the High-Quality (HQ) and Useless classifications. The resulting dataset state distributions and the downstream model performances are detailed in Table 5.

*Table 5.* Sensitivity analysis of triage scoring thresholds. Avg Winning Score: the average Winning Score compared against the FULL baseline across the WizardLM, Vicuna, Koala, LIMA, and Sinstruct evaluation datasets.

| Setting | HQ Rule | Useless Rule | HQ (%) | Fixable (%) | Useless (%) | Avg Winning Score |
|---|---|---|---|---|---|---|
| HQ LOOSER | 4,4,4,3 | 4,3,3 | 20.7 | 66.0 | 13.3 | 1.78 |
| HQ STRICTER | 5,4,5,3 | 4,3,3 | 17.1 | 69.6 | 13.3 | 1.75 |
| USELESS LOOSER | 5,4,4,3 | 4,3,2 | 20.0 | 67.6 | 12.4 | 1.80 |
| USELESS STRICTER | 5,4,4,3 | 4,3,4 | 20.0 | 60.7 | 19.3 | 1.78 |
| Default | 5,4,4,3 | 4,3,3 | 20.0 | 66.6 | 13.4 | 1.78 |

The results indicate that Cure-SFT is fairly robust to moderate changes in the triage thresholds. As expected, modifying the strictness of the rules alters the state distribution of the dataset. For instance, a stricter HQ rule reduces the proportion of High-Quality samples from 20.0% to 17.1%, which correspondingly increases the size of the Fixable pool. Conversely, a stricter Useless rule increases the discard rate from 13.4% to 19.3%.

Despite these distributional changes, the downstream performance remains stable. The Average Winning Score varies between 1.75 and 1.80, which is comparable to the 1.78 obtained with the default setting. This suggests that the effectiveness of Cure-SFT is not strictly dependent on heavily tuned thresholds, but rather relies on the stage-wise separation of semantic redundancy and quality defects, as well as the targeted remediation strategy.

## E.2. Evaluation on the Mistral-7B backbone

In addition to LLaMA-3-8B, we conduct instruction fine-tuning using Mistral-7B as the base model, following the same setting as described in the main experiments. As shown in Table 6, Cure-SFT maintains its performance advantage when applied to Mistral-7B. These results indicate that the benefits of Cure-SFT generalize across different LLM architectures.

## E.3. Controlled Ablation on the Fixable Subset

To isolate the effect of the remediation strategy, we conduct a controlled ablation study on the *Fixable* subset identified from the Alpaca dataset. In this setup, fine-tuning is strictly restricted to this identical subset. We compare two variants:

- **Unguided Rewrite**: rewriting all samples in the Fixable subset without diagnostic guidance.

- **Targeted Remediation**: applying diagnosis-guided remediation to the same Fixable subset.

The evaluation results of training on these two processed subsets are detailed in Table 7.

The results show that Targeted Remediation achieves an average score of 1.68, compared to 1.51 for the Unguided Rewrite baseline. This performance difference suggests that utilizing explicit diagnostic signals provides critical guidance for data refinement, yielding better downstream performance than free-form text regeneration.

*Table 6.* Evaluation results on the Mistral-7B backbone. Higher scores indicate better performance. Best results are shown in **bold**.

| Dataset | Method | Winning Score (vs. FULL) | | | | | | AlpacaEval 2.0 | |
|---|---|---|---|---|---|---|---|---|---|
| | | WizardLM | Vicuna | Koala | LIMA | Sinstruct | Avg. | LC Win | Win |
| Alpaca (10%) | Random | 0.95 | 1.01 | 1.01 | 0.97 | 0.95 | 0.98 | **12.58** | 4.57 |
| | IFD | 1.29 | 1.42 | 1.20 | 1.38 | 1.10 | 1.28 | 8.89 | 4.05 |
| | Deita | 1.04 | 1.26 | 1.13 | 1.13 | 1.02 | 1.12 | 9.99 | 4.21 |
| | Alpagasus | 1.10 | 1.19 | 0.98 | 1.17 | 0.99 | 1.09 | 9.78 | 4.00 |
| | T-SHIRT | 0.93 | 1.24 | 1.04 | 1.01 | 0.85 | 1.01 | 10.86 | 4.40 |
| | Rewrite | 1.69 | 1.84 | 1.43 | 1.70 | 1.23 | 1.58 | 12.56 | 6.37 |
| | CoachLM | 1.23 | 1.28 | 1.12 | 1.09 | 1.08 | 1.16 | 9.91 | 4.08 |
| | Selective-RT | 1.60 | **1.90** | 1.67 | 1.83 | 1.48 | 1.70 | 5.28 | 3.92 |
| | NILE | 1.62 | 1.84 | 1.71 | 1.85 | 1.56 | 1.72 | 11.84 | 6.45 |
| | **Cure-SFT** | **1.79** | 1.87 | **1.78** | **1.86** | **1.61** | **1.78** | 10.42 | **6.68** |
| WizardLM (10%) | Random | 0.89 | 0.91 | 0.86 | 0.98 | 0.76 | 0.88 | 10.84 | 4.55 |
| | IFD | 0.88 | 0.93 | 0.89 | 0.94 | 0.86 | 0.90 | 9.57 | 4.35 |
| | Deita | 0.99 | 0.93 | 1.03 | 1.07 | 1.00 | 1.00 | 9.59 | 5.00 |
| | Alpagasus | 0.93 | 0.85 | 0.91 | 0.93 | 0.85 | 0.89 | 8.41 | 3.88 |
| | T-SHIRT | 0.86 | 0.96 | 0.93 | 0.93 | 0.85 | 0.91 | **12.24** | 4.99 |
| | Rewrite | 1.02 | 1.23 | 1.00 | 1.36 | 0.90 | 1.10 | 11.45 | 5.68 |
| | CoachLM | 0.90 | 0.97 | 1.03 | 1.08 | 0.92 | 0.98 | 10.37 | 5.13 |
| | Selective-RT | 1.17 | 1.23 | 1.25 | **1.38** | 1.19 | 1.24 | 4.74 | 3.83 |
| | NILE | 1.13 | 1.26 | 1.24 | 1.37 | 1.20 | 1.24 | 11.98 | 5.79 |
| | **Cure-SFT** | **1.25** | **1.28** | **1.28** | 1.36 | **1.25** | **1.28** | 10.69 | **6.04** |

*Table 7.* Controlled ablation on the Fixable subset from the Alpaca dataset. Both methods are applied to the identical subset.

| Method | WizardLM | Vicuna | Koala | LIMA | SinStruct | Avg |
|---|---|---|---|---|---|---|
| Unguided Rewrite | 1.58 | 1.73 | 1.43 | 1.65 | 1.18 | 1.51 |
| Targeted Remediation | **1.69** | **1.88** | **1.63** | **1.78** | **1.44** | **1.68** |

