# OpenReview forum: "Cure-SFT: Diagnostic-Guided Data Curation for Instruction Tuning"
_ICML.cc/2026/Conference — ICML 2026 regular_

### Official Review · Reviewer_udmw · 2026-02-27

**Soundness:** 2
**Presentation:** 3
**Significance:** 3
**Originality:** 2
**Overall Recommendation:** 4
**Confidence:** 4

**Summary:**

The existing data curation work is hard to simultaneously maintain data quality, diversity, and distributional consistency. This work introduce Cure-SFT, a coarse-to-fine, diagnostic-guided method for instruction data curation that explicitly disentangles semantic redundancy from quality defects. The experiment results show that Cure-SFT surpass the strong selection-based and rewriting-based methods.

**Compliance With Llm Reviewing Policy:**

Affirmed.

**Final Justification:**

Since most of my questions have been addressed, I adjust the score accordingly.

**Key Questions For Authors:**

1. What's the time consumption for each stage, and what's the cost comparing to training time?

2. It seems the most benefit come from the third stage according to Table 2?

**Limitations:**

yes

**Strengths And Weaknesses:**

# Strengths:
1. It combines the benefits of selection-based and rewriting-based methods. Unlike the traditional keep-or-drop approach, it defines three states for each data point, they are high-quality, fixable, and useless.

2. It enhances model training efficiency, achieving comparable results while using only 10% of the budget. Meanwhile, it seems a robust method across different scale and quality datasets.

# Weaknesses:
1. Time Consumption Issue: While training costs are significantly reduced, the three-stage processing (embedding, cluster, LLM for diagnosis, LLM for remediation) introduces substantial additional overhead compared to direct data selection methods.

2. In the first stage, a uniform sampling strategy is applied across all clusters to prevent dominance by large or high-frequency semantic clusters. But this might discard high-quality samples and retain similar, less useful ones, it's more effective to perform deduplication after establishing a scoring system or criteria.

3. The three stages—data deduplication, data filtering, and data rewriting—are part of a standard data curation pipeline. The second stage introduces a three-state approach instead of the usual two, but overall, it's challenging to find novelty in the pipeline.

---

> ### Author Rebuttal · Authors · 2026-03-30
>
> ***Q1: Time Consumption Issue***
>
> **A1:** We thank the reviewer for raising this important concern. To quantify this issue, **R4-Table 1** reports a token- and pass-level estimate of curation cost on **Alpaca**.
>
> **R4-Table 1: Data curation costs on Alpaca.**
> |Method|Full-dataset LLM processing?|Teacher / Scoring Calls|Input Tokens|Output Tokens|Key Characteristics|
> |-|-|-|-|-|-|
> |IFD|Yes|104,004|~6.7M|0|Two full-dataset passes for IFD scoring.|
> |AlpaGasus|Yes|52,002|~10.4M|~1.3M–4.2M|Evaluates every sample using a prompt.|
> |Deita|Yes|104,004|~7.2M|~0.1M|Requires two full-dataset scorer passes.|
> |Rewrite|No|5,200|~0.3M|~0.8M|Rewrites the randomly sampled 10% without diagnosis.|
> |Cure-SFT|No|8,665|~3.1M|~0.5M–0.8M|Diagnosis on 5.2k candidates; remediation on 3.4k Fixable samples.|
>
> As shown in **R4-Table 1**, Cure-SFT does **not** incur higher overall curation cost than representative **full-dataset selection baselines** such as **IFD, Deita, and AlpaGasus**. The reason is that its teacher-based stages are applied only to a much smaller subset rather than the full dataset. In particular, **Stage I** is a lightweight one-time preprocessing step that takes about **6 minutes on a single RTX 5090** in our Alpaca setup, reducing the full dataset to about **5.2K** candidates. **Stage II** is then applied only to these candidates, and **Stage III** only to about **3.5K Fixable** samples. Thus, Cure-SFT trades full-dataset scoring for selective diagnosis and targeted remediation rather than introducing substantial full-dataset overhead.
>
> ---
>
> ***Q2: It's more effective to perform deduplication after establishing a scoring system or criteria.***
>
> **A2:** We thank the reviewer for this thoughtful comment. The alternative of **scoring first, then deduplicating** is indeed intuitive, but our choice to **deduplicate first, then score** is deliberate.
>
> First, if quality scoring were applied first, some semantic categories could be reduced too early, causing an irreversible loss of coverage. Cure-SFT is designed to avoid this by preserving coverage first and deferring quality handling to **Stage II/III**, where imperfect but informative samples can still be diagnosed and remediated. This allows the pipeline to better balance semantic coverage and data quality, rather than sacrificing one too early for the other.
>
>  Second, this design is also more efficient. If fine-grained teacher-based scoring were applied before deduplication, it would need to run on the full dataset. In contrast, our coarse-to-fine pipeline first compresses the data, then applies diagnosis and remediation only to the reduced candidate set.
>
> In summary, our design does not assume that uniform sampling is quality-optimal by itself. Rather, it reflects the view that **semantic coverage is handled first, while instance-level quality is assessed and repaired in later stages**.
>
> ---
>
> ***Q3: It's challenging to find novelty in the pipeline.***
>
> **A3:** We thank the reviewer for this important comment. We agree that **deduplication, filtering, and rewriting are common operations when viewed in isolation**. Our novelty claim is therefore **not** that Cure-SFT is simply a new ordering of standard modules.
>
> Rather, the key novelty lies in **how the problem is decomposed**. Cure-SFT explicitly distinguishes **semantic redundancy** from **quality defects**, and treats them in different stages. In contrast to more conventional pipelines that let early quality control determine what is retained, our method first preserves **broad semantic coverage** through redundancy reduction in **Stage I**, and only then applies **quality diagnosis and correction** in **Stage II/III**. We view this as a different curation philosophy: semantic coverage is handled first, while quality issues are addressed downstream rather than being allowed to collapse coverage prematurely.
>
> Besides, the main methodological novelty is then the coupling between **Stage II** and **Stage III** through an explicit intermediate Fixable state. Stage II does not perform a binary keep/discard filter; it outputs **High-Quality / Fixable / Useless** triage together with a diagnostic signal. The Fixable state is not just an extra label—it is the interface that enables **targeted remediation** only for samples that are semantically valid but locally flawed. This also makes **Stage III** different from standard rewriting baselines: it is a **diagnosis-conditioned, minimal-intervention repair operator**, rather than holistic rewriting.
>
> Our ablation suggests that this is more than a superficial pipeline assembly. On Alpaca, **Stage I alone** achieves **1.05**, **Stage I+II** reaches **1.19**, **Stage II+III** already reaches **1.72**, and the **full method** reaches **1.78**. This pattern suggests that the main gain comes from the structured interaction between **coverage preservation, diagnostic triage, and targeted remediation**, rather than from simply stacking common modules.

---

> > ### Author Rebuttal · Reviewer_udmw · 2026-04-04
> >
> > Thank you to the authors for your response. Most of my questions have been addressed, and I will adjust the score accordingly.

---

> > > ### Author Response · Authors · 2026-04-07
> > >
> > > Thank you very much for your continued engagement and for updating your score. We are glad that our responses have addressed most of your questions. Your insightful comments have helped us improve the clarity and rigor of our work.

---

### Official Review · Reviewer_BUxm · 2026-03-11

**Soundness:** 3
**Presentation:** 3
**Significance:** 3
**Originality:** 3
**Overall Recommendation:** 4
**Confidence:** 4

**Summary:**

This paper proposes Cure-SFT, a coarse-to-fine data curation pipeline for instruction tuning of large language models. The motivation is that existing data curation approaches—primarily selection-based filtering and rewriting-based correction—often fail to distinguish between semantic redundancy and localized quality defects in instruction datasets. To address this, Cure-SFT introduces a three-stage pipeline. First, the method performs stratified semantic–geometric sampling in the embedding space to remove redundant samples while preserving semantic diversity. Second, a teacher model performs diagnostic triage, categorizing samples into three states: High Quality, Fixable, or Useless. Third, the pipeline applies targeted remediation only to samples identified as Fixable, while preserving high-quality samples and discarding unusable ones. Experimental results show that models fine-tuned on 10% of the curated data using Cure-SFT outperform models trained on the full dataset or on subsets constructed by several strong baselines, suggesting improvements in both data quality and diversity preservation.

**Compliance With Llm Reviewing Policy:**

Affirmed.

**Key Questions For Authors:**

Questions

1. How sensitive is the method to the scoring thresholds used in the triage stage? For example, if the thresholds for labeling samples as High Quality, Fixable, or Useless change, does the final model performance change significantly?

2. Have the authors tested the curated datasets with other model families besides LLaMA-3-8B? It would be useful to know if the improvements hold across different architectures or model sizes.

3. What is the computational cost of running the Cure-SFT pipeline? Since a 72B teacher model is used for triage and rewriting, how does the cost compare to baseline data curation methods?

4. How well does the method perform when using larger portions of the dataset? The biggest gains appear at 10% data—do similar improvements hold when training with more data?

**Limitations:**

No. The authors include a brief "Impact Statement" regarding societal implications, but they do not adequately discuss the technical limitations of their work.

Here are constructive suggestions for improvement:

1. Discuss the computational cost and time required to run the data curation pipeline.
2. Analyze how sensitive the final results are to the specific scoring rules and thresholds used.
3. Discuss whether the benefits of this method apply to other types of language models.
4. Address how the method's effectiveness changes when applied to much larger datasets.

**Strengths And Weaknesses:**

Strengths

1. The paper identifies a meaningful limitation of existing data curation methods for instruction tuning, the inability to distinguish between semantic redundancy and localized quality defects. The proposed three-state categorization (High Quality, Fixable, Useless) provides a more nuanced view of instruction data quality compared to binary filtering approaches.
2. By applying rewriting only to samples classified as Fixable while preserving high-quality data, the method attempts to avoid the style homogenization often introduced by large-scale rewriting pipelines. This design choice is well motivated and aligns with the goal of maintaining diversity in instruction datasets.
3. The evaluation spans multiple benchmarks and two instruction datasets, and includes ablations that remove individual components of the pipeline. These experiments provide evidence that the different stages of the pipeline contribute to the overall performance improvements.
4. The paper is generally well structured, with clear diagrams illustrating the pipeline and detailed descriptions of prompts and training settings, which improves the reproducibility of the experiments.

Weaknesses

1. The triage stage relies on hard thresholds (e.g., a score of 5 for instruction compliance to classify samples as High Quality). The paper does not provide an analysis of how sensitive the results are to these threshold choices, which raises questions about robustness.
2. All training experiments are conducted using LLaMA-3-8B. It is unclear whether the curated datasets would provide similar benefits when fine-tuning models from different architectures or parameter scales.
3. The experimental results suggest that improvements are most pronounced when using small data budgets (e.g., 10%), while larger budgets yield diminishing returns. This raises the question of how broadly the method benefits full-scale instruction tuning scenarios.
4. The diagnostic and rewriting stages rely on a 72B-parameter teacher model, yet the paper does not provide a comparison of the computational cost or latency relative to baseline data curation approaches. This omission makes it difficult to evaluate the practical efficiency of the proposed method.

---

> ### Author Rebuttal · Authors · 2026-03-30
>
> ***Q1: How sensitive is the method to the scoring thresholds used in the triage stage?***
>
> **A1:** We thank the reviewer for this important question. We agree that the sensitivity to the triage thresholds should be clarified.
>
> To directly test sensitivity, we varied both the **High-Quality** and **Useless** rules around the default setting:
>
> **R3-Table 1: Sensitivity analysis of triage scoring thresholds.**
>
> |**Setting**|**HQ Rule**|**Useless Rule**|**HQ (%)**|**Fixable (%)**|**Useless (%)**|**Avg Winning Score**|
> |-|-|-|-|-|-|-|
> |HQ LOOSER|4,4,4,3|4,3,3|20.7%|66.0%|13.3%|1.78|
> |HQ STRICTER|5,4,5,3|4,3,3|17.1%|69.6%|13.3%|1.75|
> |USELESS LOOSER|5,4,4,3|4,3,2|20.0%|67.6%|12.4% |1.80|
> |USELESS STRICTER |5,4,4,3| 4,3,4|20.0%|60.7%|19.3%|1.78|
> |Ours|5,4,4,3|4,3,3| 20.0%|66.6%|13.4%|1.78|
>
> The results show that Cure-SFT is fairly robust to moderate threshold changes. Although the proportions of High-Quality / Fixable / Useless vary somewhat, the final average winning score remains very stable (1.75–1.80, vs. 1.78 for the default setting). This suggests that the method does not depend critically on a finely tuned threshold choice.
>
> ---
>
> ***Q2: Have the authors tested the curated datasets with other model families besides LLaMA-3-8B?***
>
> **A2:** We thank the reviewer for this helpful suggestion. Please see our response to **Reviewer Zfdj, Q1**, where we report additional experiments on **Mistral-7B** under the same curation and evaluation protocol. In brief, the relative trend remains consistent: Cure-SFT still achieves the best overall average winning score on both **Alpaca** and **WizardLM**, suggesting that the improvement is not specific to the original **LLaMA-3-8B** setting.
>
> ---
>
> ***Q3: What is the computational cost of running the Cure-SFT pipeline?***
>
> **A3:** We thank the reviewer for this important question. Please see our response to **Reviewer udmw, Q1**, where we provide a detailed cost analysis. In brief, although Cure-SFT uses a **72B teacher** for triage and remediation, its teacher-based stages are applied only to a much smaller subset rather than to the full dataset. As a result, Cure-SFT requires only about **8.7K** teacher/scoring passes on **Alpaca**, compared with **52K** for **AlpaGasus** and **104K** for **IFD/DEITA**. A token-level estimate shows the same trend: Cure-SFT uses about **3.1M** input tokens and **0.5M–0.8M** output tokens, which is lower than representative full-dataset selection baselines.
>
> ---
>
> ***Q4: How well does the method perform when using larger portions of the dataset? The biggest gains appear at 10% data—do similar improvements hold when training with more data?***
>
> **A4:** We thank the reviewer. We agree that the gains are most pronounced at the **10%** budget, but Cure-SFT is not limited to this setting. For instance, increasing the budget to **15%** still yields a strong average winning score of **1.78** on the **WizardLM** test set, showing that the method remains effective beyond the 10% setting.
>
> This trend is consistent with our **stage-wise design**. At larger budgets, the marginal value of redundancy reduction (**Stage I**) naturally decreases as the retained set already covers more of the semantic space. However, Cure-SFT is not only a Stage-I method: **Stage II** and **Stage III** still provide explicit quality control by separating High-Quality / Fixable / Useless samples and improving the Fixable subset. This also helps explain why Cure-SFT is not limited to the 10% setting: its later stages still provide explicit quality control over the retained data.
>
> More broadly, this behavior is also consistent with the main goal of Cure-SFT. Our method is primarily aimed at the data-efficient regime, where the key question is whether a smaller, better-curated dataset can rival or outperform a larger, noisier one, rather than whether curation can continue to produce equally large gains when nearly the full dataset is already retained. This perspective is also consistent with LIMA [1], which shows that a small, carefully curated set of high-quality instruction data can already induce strong instruction-following ability.
>
> [1] Zhou et al., LIMA: Less is More for Alignment, NeurIPS 2023.

---

> > ### Author Rebuttal · Reviewer_BUxm · 2026-04-03
> >
> > I thank the authors for their thorough rebuttal, which successfully resolved my primary technical concerns. I am keeping my score of Weak Accept, as I feel this rating is sufficiently high and accurately reflects the paper's contributions.

---

> > > ### Author Response · Authors · 2026-04-07
> > >
> > > Thank you very much for your thoughtful feedback and for acknowledging our rebuttal. We truly appreciate your careful evaluation and are glad that the rebuttal has successfully addressed your primary technical concerns.

---

### Official Review · Reviewer_Zfdj · 2026-03-12

**Soundness:** 3
**Presentation:** 3
**Significance:** 3
**Originality:** 3
**Overall Recommendation:** 5
**Confidence:** 4

**Summary:**

This paper proposes a three-stage pipeline for curating instruction-tuning training data. It 1) selects semantically diverse data, 2) uses a 72B-parameter teacher model to partition data into high-quality, fixable, and useless data, and 3) uses that same teacher model to rewrite fixable data. The rewritten data and the original high-quality data are together used as the final training set. The authors evaluate the method and baselines by training LLaMA-3-8B on 10% subsets of two datasets (Alpaca and WizardLM), evaluating on five benchmarks, and comparing win rates against models trained on the full dataset. The model convincingly outperforms baselines in this setting.

**Compliance With Llm Reviewing Policy:**

Affirmed.

**Final Justification:**

The authors have addressed my concerns by evaluating on an extra model and performing additional ablations that train on the same fixable subset.

**Key Questions For Authors:**

1. I'm still not entirely sure where in your pipeline the extra performance is coming from. I'd love to see further ablations that might further disentangle the effect of rewriting on the data you choose (since the rewrite baseline randomly samples examples), such as: a) an ablation that only uses the rewritten fixable examples without the high-quality examples, or even b) an ablation that rewrites the high-quality examples.

2. How do you think your method would compare to other baselines that also prioritize diversity like G-Vendi [1]?

Small things:
- I really like that Figure 1 explains your method very well. I think it would be even clearer if the teacher model is also incorporated.
- It takes a long time to load some pages in the pdf. The images seem like they might be very high-resolution pngs right now--if so, you could change them to pdfs.

[1]: Jung et al., Prismatic Synthesis: Gradient-based Data Diversification Boosts Generalization in LLM Reasoning, NeurIPS 2025.

**Limitations:**

yes

**Strengths And Weaknesses:**

Strengths:
1. The proposed method is very clearly described and convincingly outperforms baselines.
2. Sections 4.5 and 4.6 contain very useful ablations and analyses.


Weaknesses:
1. I would have liked to see more models used for training in the evaluation. Currently only a single model (LLaMA-3-8B) is used.
2. Contribution 1 in the final paragraph of the introduction does not seem explicitly supported by any experiments ("(1) We identify a common limitation in existing instruction data curation methods: they fail to distinguish semantic redundancy from quality defects and overlook the inherent three state quality structure of instruction data.")

---

> ### Author Rebuttal · Authors · 2026-03-30
>
> ***Q1: I would have liked to see more models used for training in the evaluation. Currently only a single model (LLaMA-3-8B) is used.***
>
> **A1:** We thank the reviewer. To address this, we additionally evaluated on a different model family, **Mistral-7B**.
>
> **R2-Table1 : Evaluation results on the Mistral-7B backbone.**
> |**Dataset**|**Method**|**WizardLM**|**Vicuna**|**Koala**|**LIMA**|**SinStruct**|**Avg**|**LC Win**|**Win**|
> |-|-|-|-|-|-|-|-|-|-|
> |**Alpaca**|Random|0.95|1.01|1.01|0.97|0.95|0.98|**12.58**|4.57|
> ||IFD|1.29|1.42|1.20|1.38|1.10|1.28|8.89|4.05|
> ||Deita|1.04|1.26|1.13|1.13|1.02|1.12|9.99|4.21|
> ||Alpagasus|1.10|1.19|0.98|1.17|0.99|1.09|9.78|4.00|
> || Rewrite|1.69|1.84|1.43|1.70|1.23|1.58|12.56|6.37|
> ||**Cure-SFT**|**1.79**|**1.87**|**1.78**|**1.86**|**1.61**|**1.78**|10.42|**6.68**|
> |**WizardLM**|Random|0.89|0.91|0.86|0.98|0.76|0.88|10.84 |4.55|
> ||IFD|0.88|0.93|0.89|0.94|0.86|0.90|9.57|4.35|
> ||Deita|0.99|0.93|1.03|1.07|1.00|1.00|9.59|5.00|
> ||Alpagasus|0.93|0.85|0.91|0.93|0.85|0.89|8.41|3.88|
> ||Rewrite|1.02|1.23|1.00|1.36|0.90|1.10|**11.45**|5.68|
> ||**Cure-SFT**|**1.25**|**1.28**|**1.28**|**1.36**|**1.25**|**1.28**|10.69|**6.04**|
>
> Cure-SFT achieves the best average winning scores. This shows that the relative advantage of Cure-SFT is preserved beyond the original LLaMA-3-8B setting. As a data-centric curation pipeline, Cure-SFT inherently provides model-agnostic benefits that explain this consistent trend.
>
> ---
>
> ***Q2: Contribution 1 in the final paragraph of the introduction does not seem explicitly supported by any experiments.***
>
> **A2:** We thank the reviewer for this observation. We agree that the current wording of Contribution 1 is too strong and may read like a standalone experimental claim. We will revise it as a diagnostic observation supported by the following analyses:
>
> **1: Separation of semantic redundancy vs. quality defects.**
> - **Pipeline analysis:** Selection-based methods mainly perform keep/discard decisions, while rewriting-based methods apply holistic rewriting.
> - **Figure 3:** **AlpaGasus** produces a more compact semantic distribution with visible gaps, whereas **Cure-SFT** preserves coverage much closer to the original distribution.
> - **Table 2:** Removing either semantic stratification or diagnosis/remediation degrades performance, showing that redundancy control and quality correction contribute separately.
>
> **2: Three-state quality structure of instruction data.**
> - **Figure 2:** **Fixable** is the largest category on both datasets.
> - **Table 2:** Removing **Stage III** and retaining only **High-Quality** samples reduces the average Winning Score from **1.78 to 1.19**, showing that a binary usable/unusable view loses substantial training value.
> - **Figure 4:** Indiscriminate rewriting causes larger distributional shift, whereas **Cure-SFT** preserves more of the original structure by leaving high-quality samples unchanged and correcting only the **Fixable** subset.
>
> ---
>
> ***Q3: I'd love to see further ablations that might further disentangle the effect of rewriting on the data you choose.***
>
> **A3:** We thank the reviewer for this very helpful suggestion. To address this point, please also see our response to **Reviewer R9Dc, Q2.1**, where we report a controlled ablation on Alpaca with training on the same Fixable subset.
>
> In that experiment, Unguided Rewrite achieves 1.51 Avg, while Targeted Remediation improves this to 1.68, showing that the gain does not come from rewriting alone, but from diagnostic-guided correction of specific defects. The full Cure-SFT further reaches 1.78, indicating that the benefit is not only from remediating Fixable samples, but also from combining them with preserved High-Quality samples.
>
> This helps disentangle the source of the gain more clearly: (i) Targeted Remediation is stronger than Unguided Rewriting on the same Fixable subset, and (ii) retaining High-Quality samples yields further gains.
>
> ---
>
> ***Q4: How do you think your method would compare to other baselines that also prioritize diversity like G-Vendi [1]?***
>
> **A4:** We thank the reviewer for highlighting G-Vendi, a strong diversity-oriented baseline. A key difference is that G-Vendi expands diversity through data synthesis, whereas Cure-SFT curates an existing dataset. More importantly, G-Vendi improves which regions are covered, but does not directly provide a mechanism to repair informative but imperfect samples. In contrast, Cure-SFT explicitly models three states and applies targeted remediation to the Fixable subset. We therefore view G-Vendi as a strong comparator for the diversity aspect of the problem, while Cure-SFT contributes an additional mechanism—quality-state modeling and targeted remediation—beyond diversity-oriented generation alone.
>
> ---
>
> ***Q5 & Q6: Minor suggestions on Figure 1 and PDF loading.***
>
> **A5 & A6:** We appreciate the suggestions and will add the teacher model to Figure 1 and use vector PDFs for faster loading in the final revision.

---

> > ### Author Rebuttal · Reviewer_Zfdj · 2026-04-04
> >
> > My concerns are resolved. Thanks for running the additional experiments and ablations.

---

> > > ### Author Response · Authors · 2026-04-07
> > >
> > > Thank you very much for your thoughtful feedback. We truly appreciate your careful review and constructive suggestions. We are glad that the additional experiments and ablations have addressed your concerns.

---

### Official Review · Reviewer_R9Dc · 2026-03-13

**Soundness:** 3
**Presentation:** 3
**Significance:** 3
**Originality:** 3
**Overall Recommendation:** 4
**Confidence:** 3

**Summary:**

This paper points out that previous instruction data curation methods often fail to distinguish between semantic redundancy and quality defects, and typically treat data in a binary manner as either usable or unusable. Instead, the authors categorize data quality into three states: high-quality, fixable, and useless. They then propose Cure-SFT, which selectively remediates fixable samples based on diagnostic assessments from a teacher model.

**Compliance With Llm Reviewing Policy:**

Affirmed.

**Final Justification:**

Although the authors did not fully address all concerns, they addressed what I consider the main weakness, so I will raise my score to 4.

**Key Questions For Authors:**

Please refer to the weaknesses.

**Limitations:**

yes

**Strengths And Weaknesses:**

- The main strength of this paper is that it does not overlook fixable samples, which are often ignored in previous approaches. It effectively refines them through a diagnostically guided targeted remediation strategy, thereby producing high-quality instruction data.

- The experimental validation is somewhat limited, particularly due to the lack of comparisons with various recent studies. Therefore, including more baselines from both selection-based and rewriting-based methods would more clearly demonstrate the superiority of the proposed approach.

- In particular, Table 2 suggests that the main performance gains come from data rewriting with a strong teacher model. However, since the comparison is mostly limited to a simple uniform rewriting baseline, it is difficult to tell whether the proposed method is actually superior to other rewriting-based approaches.

---

> ### Author Rebuttal · Authors · 2026-03-30
>
> ***Q1: Including more baselines from both selection-based and rewriting-based methods.***
>
> **A1:** We thank the reviewer for this helpful suggestion. To address this concern, we added two recent baselines: **T-SHIRT** [1], a **selection-based** method based on token-selective hierarchical data selection, and **Selective-RT** [2], a **rewriting-based** method that combines reflection-based rewriting with student-side selection.
>
> **R1-Table 1: Performance comparison of Cure-SFT against T-SHIRT and Selective-RT.**
>
> |**Dataset**| **Method**|**WizardLM**| **Vicuna**|**Koala**|**LIMA**|**SinStruct**|**Avg**|**LC Win**|Win|
> |-|-|-|-|-|-|-|-|-|-|
> |**Alpaca**|T-SHIRT|1.01|1.36|1.09|1.01|0.86|1.07|9.90|3.95|
> ||Selective-RT|1.63|1.85|1.52|1.74|1.39|1.63|4.08|3.48|
> || **Cure-SFT**|**1.85**|**1.91**|**1.78**|**1.83**|**1.52**|**1.78**|**11.45**|**7.66**|
> |**WizardLM**|T-SHIRT|0.82|1.01|0.89|0.84|0.82|0.88|9.15|4.13|
> ||Selective-RT|1.15|1.25|1.22|**1.32**|1.16|1.22|4.81|4.37|
> ||**Cure-SFT**|**1.32**|**1.26**|**1.34**|1.26|**1.19**|**1.27**|**9.67**|**7.24**|
>
> These new comparisons are consistent with our main conclusion. Cure-SFT remains clearly stronger than **T-SHIRT** and also outperforms **Selective-RT**. We believe the reason is that these baselines capture only part of the problem addressed by Cure-SFT. **T-SHIRT** improves data selection, but like other selection-based methods, it can only keep or discard samples and cannot repair fixable ones. **Selective-RT** is a stronger rewriting-based baseline that combines reflection-based rewriting with student-side selection. However, unlike Cure-SFT, it does not explicitly model different quality states of samples, but instead lets the student choose between the original and a teacher-reflected version based on r-IFD. In addition, its reflection step generates a new complete instruction or response, which is still essentially a holistic rewrite. By contrast, Cure-SFT explicitly identifies a Fixable subset and applies diagnosis-guided targeted remediation that focuses on correcting localized defects while preserving the original valid semantic structure. We believe this more selective handling of imperfect samples better explains the advantage of **Cure-SFT** over **Selective-RT**.
>
> [1] Fu et al., T-SHIRT: Token-Selective Hierarchical Data Selection
> for Instruction Tuning, NeurIPS 2025.
>
> [2] Li ei al., Selective Reflection-Tuning: Student-Selected Data Recycling for LLM Instruction-Tuning, ACL 2024.
>
> ---
>
> ***Q2.1: Table 2 suggests that the main performance gains come from data rewriting with a strong teacher model.***
>
> **A2.1:** We thank the reviewer for this insightful comment. While Table 2 highlights the importance of remediation, Cure-SFT is not a generic rewriting method. Unlike holistic rewriting approaches that revise all selected samples, Cure-SFT rewrites only the Fixable subset and does so through diagnosis-guided remediation rather than free-form rewriting.
>
> To isolate the source of the gain, we ran a controlled ablation on Alpaca where training is restricted to the same Fixable subset identified by our pipeline.
>
> **R1-Table 2: Controlled ablation on the Fixable subset.**
>
> |**Method**|**WizardLM**|**Vicuna**|**Koala**|**LIMA**|**SinStruct**|**Avg**|
> |-|-|-|-|-|-|-|
> |Unguided Rewrite|1.58|1.73|1.43|1.65|1.18|1.51|
> |**Targeted Remediation**|**1.69**|**1.88**|**1.63**|**1.78**|**1.44**|**1.68**|
>
> Here, **Unguided Rewrite** and **Targeted Remediation** are applied to the same Fixable subset. Targeted Remediation clearly outperforms Unguided Rewrite, showing that the gain does not come from rewriting alone, but from using diagnostic signals to determine what to correct and how to correct it. In addition, Section 4.6.4 shows that Cure-SFT remains effective with both 14B and 72B teachers, suggesting that the gain is not overly dependent on a single very large teacher.
>
> ---
>
> ***Q2.2: The comparison is mostly limited to a simple uniform rewriting baseline.***
>
> **A2.2:** We thank the reviewer for this important concern. As introduced in our response to **Q1**, we evaluated Cure-SFT against **Selective-RT**,  Cure-SFT still performs better overall: on Alpaca, its Avg is 1.78 vs. 1.63 for Selective-RT; on WizardLM, 1.27 vs. 1.22. This suggests that the gain is not simply from using a strong teacher to repair data.
>
> We believe this advantage comes from a key methodological difference in how imperfect samples are handled. Rewriting-based baselines such as Selective-RT revise samples through teacher-guided reflection, but the resulting update is still largely a holistic rewrite. In contrast, Cure-SFT explicitly identifies a Fixable subset and applies diagnosis-guided remediation. Rather than regenerating the full response, it focuses on correcting the diagnosed local defects while preserving the original valid semantic structure. This more selective form of correction, we believe, better explains the stronger downstream performance.

---

> > ### Author Rebuttal · Reviewer_R9Dc · 2026-04-04
> >
> > Thank you for the detailed explanation.
> >
> > I still have concerns regarding Q2-1. The additional experiment shows that Targeted Remediation improves performance, but the size of the gain does not seem very large. For example, in Table 2, removing Stage 3 gives an average score of 1.19, while unguided rewriting alone already increases it to 1.51. This suggests that much of the gain comes from rewriting itself, while the added benefit of Targeted Remediation appears relatively modest.
> >
> > This leads to my concern in Q2.2. I am still not fully convinced that the proposed method is clearly better than other rewriting-based approaches. Although the authors compare with Selective-RT (2024), it is unclear whether this was done under a fair experimental setting. In particular, the response does not clearly explain whether the student and teacher models were matched to those used in the proposed method, or whether Selective-RT was fairly optimized in terms of training hyperparameters. Also, Selective-RT (2024) is not a recent rewriting-based method, and it is the only rewriting-based baseline included here, so this comparison still seems insufficient.

---

> > > ### Author Response · Authors · 2026-04-07
> > >
> > > ## **Response to further concerns on Q2.1 and Q2.2**
> > > We thank the reviewer for the continued engagement. To address the concerns regarding rewriting-based baselines, we further expanded the evaluation to include two additional methods: **CoachLM** [1], which trains a dedicated model on expert revisions for automatic instruction correction, and **NILE** [2], which revises responses to better align with the model’s internal knowledge.
> > > ### **Regarding Q2.2**
> > > We group recent rewriting-based methods into two paradigms. The first comprises methods that use a frozen large LLM as the main rewriting / revision engine; **Selective-RT**, **NILE**, and our **Cure-SFT** fall into this category. For fairness, we implemented all methods in this category using the same teacher model (**Qwen2.5-72B-Instruct-AWQ**). The second paradigm trains a smaller dedicated rewriter; **CoachLM** falls into this category, and we evaluated it using the **ChatGLM2-6B** rewriter specified in the original paper. Across all methods, the downstream student model (**LLaMA-3-8B**) and the fine-tuning hyperparameters were kept the same.
> > >
> > > **R1-Table 3: Expanded comparison with recent rewriting-based baselines.**
> > > |**Dataset**|**Method**|**WizardLM**|**Vicuna**|**Koala**|**LIMA**|**SinStruct**|**Avg**|**LC Win**|**Win**|
> > > |-|-|-|-|-|-|-|-|-|-|
> > > |**Alpaca**|CoachLM|1.17|1.20|1.12|1.05|0.99 |1.11|10.32|4.23|
> > > | |Selective-RT|1.63|1.85| 1.52| 1.74     | 1.39          | 1.63     | 4.08       | 3.48     |
> > > | |NILE| 1.64         | 1.90       | 1.66      | 1.79     | 1.46| 1.69     | **11.85**  | 7.31     |
> > > | |**Cure-SFT**| **1.85**     | **1.91**   | **1.78**  | **1.83** | **1.52**      | **1.78** | 11.45      | **7.66** |
> > > | **WizardLM** |CoachLM| 0.84         | 1.01       | 0.97      | 0.96     | 0.87          | 0.93     | 7.84       | 4.76     |
> > > | |Selective-RT| 1.15         | 1.25       | 1.22      | 1.32     | 1.16          | 1.22     | 4.81       | 4.37     |
> > > | |NILE| 1.24         | **1.29**   | 1.27      | 1.22     | 1.13          | 1.23     | 9.45       | 7.14     |
> > > | |**Cure-SFT**| **1.32**     | 1.26       | **1.34**  | **1.26** | **1.19**      | **1.27** | **9.67**   | **7.24** |
> > >
> > > As shown above, Cure-SFT performs best among the evaluated rewriting-based baselines, including the recent NILE baseline.
> > >
> > > [1] Liu et al., CoachLM: Automatic Instruction Revisions Improve the Data Quality in LLM Instruction Tuning, ICDE 2024.
> > >
> > > [2] Hu et al., NILE: Internal Consistency Alignment in Large Language Models, EMNLP 2025.
> > >
> > > ### **Regarding Q2.1**
> > >
> > > We agree that unguided rewriting is already a strong contributor. However, this gain is measured on the same **Fixable** subset identified by our pipeline, so it should not be interpreted as the effect of rewriting alone: the upstream stages already contribute by isolating the subset on which rewriting is applied. Under this same data condition, **Targeted Remediation** further improves performance from **1.51 to 1.68**. At this already strong baseline level, further incremental gains become harder to obtain. In that context, the additional improvement represents a meaningful further gain from diagnosis-guided remediation.
> > >
> > > We also agree that the final performance of **Cure-SFT** reflects more than Stage III alone. In addition to targeted remediation, the overall gain also benefits from the broader **stage-wise design**, including preserving **High-Quality** samples unchanged and handling semantic redundancy separately from quality defects. We believe this advantage is partly explained by Cure-SFT’s more explicit quality-state modeling and targeted treatment of imperfect samples.
> > >
> > > More broadly, the difference from current rewriting paradigms is not only that Cure-SFT rewrites, but **how** it rewrites. Methods such as **Selective-RT**, **NILE**, and **CoachLM** still rely primarily on rewriting-style revision of the sample as a whole, which does not explicitly separate localized flaws from the valid parts of the sample. By contrast, Cure-SFT performs **diagnosis-guided targeted remediation** on the **Fixable** subset only. This allows the model to correct localized defects while better preserving the original valid semantic structure of the data.
> > >
> > > We hope that these additional experiments and clarifications help address the reviewer’s concern and make our intended claim more precise.

---

### Decision · Program_Chairs · 2026-04-30

**Decision:**

Accept (regular)

**Comment:**

This paper proposes Cure-SFT, a three-stage data curation pipeline for instruction tuning that separates redundancy reduction, quality triage, and targeted remediation. The paper is clear, the method is well motivated, and the empirical results are strong overall.

The main concern in the reviews was whether the observed gains should really be attributed to the proposed diagnostic-guided remediation rather than to rewriting more generally, as well as whether the rewriting-based comparisons were strong and fair enough. In my view, the rebuttal addressed these concerns reasonably well by adding stronger baselines, clarifying the comparison setup, and providing additional ablations and analyses. I still think the paper should be somewhat cautious in how strongly it claims to isolate the source of the gain, but this no longer seems like a decisive issue.

Overall, this is a solid empirical paper with a useful data-centric contribution to instruction tuning, and I recommend acceptance.